# UAV-Based Privacy-Preserved Trustworthy Seamless Service Agility for NextG Cellular Networks

**DOI:** 10.3390/s22072756

**Published:** 2022-04-02

**Authors:** Mai A. Abdel-Malek, Muhammad M. Sayed, Mohamed Azab

**Affiliations:** 1Department of Electrical and Computer Engineering, Virginia Tech, Blacksburg, VA 24061, USA; mmalek@vt.edu; 2Department of Computer and Communications Engineering, Alexandria University, Alexandria 21526, Egypt; muhammad.magdy@mena.vt.edu; 3Department of Computer and Information Sciences, Virgina Military Institute, Lexington, VA 24450, USA

**Keywords:** service agility, authentication, secure wireless communications, 5G and NextG, privacy, UAV assisted trust and porting

## Abstract

Next Generation cellular networks are expected to offer better service quality, secure and reliable service provisioning, and more cooperative operation even in unexpected stressful situations. Service provider cooperation can facilitate reliable service provisioning and extended coverage in disasters situations or partial network failures. However, the current 4G and 5G standards do not offer security and privacy-friendly support for inter-operator agility and service mobility, a key enabler for such cooperation. The situation becomes more critical in presence of attackers, where establishing trust relationships becomes very complicated. This paper presents a novel UAV-assisted user-agility support framework that enables trustworthy seamless service migration in a zero-trust environment. The proposed framework facilitates temporal authentication-authority delegation and proxying to enable preservice, all-party mutual authentication. The framework is implemented and tested on top of the srsRAN open-source 4G/5G software stack. Experiments showed that the presented framework managed to facilitate effective and efficient trustworthy service migration between heterogeneous service provider networks.

## 1. Introduction

Literature in the area of Next Generation (NextG) wireless communications demonstrated the desperate need for more bandwidth, enhanced connection reliability, and security. NextG networks should offer a much better user experience enabling extended coverage and seamless mobility. Accordingly, the NextG cellular infrastructures are expected to serve various heterogeneous end-users with a large set of dynamically changing requirements, capabilities, and constraints. Given the high mobility nature of such devices, such as Unmanned Aerial Vehicles (UAVs), it would be a waste not to take advantage of such devices’ existence to support the infrastructure operation in case of need.

In situations such as white or red spots, the infrastructure fails to serve the users in need. A white spot describes a tower failure due to intentional (e.g., terrorist attack) or unintentional (e.g., a natural disaster) causes. A red spot describes an unexpected excessive traffic demand that overwhelms the cellular tower and reduces service quality. Red spots can easily happen in areas where big events are happening (e.g., game days or celebrations). In the case of white spots, the situation becomes critical and life-threatening. In disaster situations, cellular service can be a key to saving many lives. In such situations, expecting user interactions to obtain better service or even basic coverage might not be realistic due to the high stress entailed or the inability to move. Users must have adequate coverage to ask for help or provide help. In white or red spots, the infrastructure endpoints represented by the cellular towers might be either overwhelmed or disabled. Flying UAV-carried towers are considered to be one of the best possible solutions to provide coverage in white spots [1,2]. Furthermore, introducing service agility where service providers can accommodate users from other networks can provide a suitable alternative for infrastructure damage or unavailability.

One of the many new features designated to the NextG standard over former generations’ standards is the service-oriented architecture that relies on network slicing and virtual network functions [3]. The network slicing provides traffic isolation functionality and supports network flexibility and service agility [4]. In this work, we propose an extended feature to be added to the 5G structure that will take the NextG service agility to the next level of seamless communication.

We propose inter-operator service agility that blends service providers into one logical network with an enhanced low latency communication performance. All the base stations (gNodeBs and eNodeBs) from different service providers are expected to provide service to users from different home networks to enhance the end-user experience. This added feature also permits seamless communication with inter-operator agility cooperative cellular environment, which provides reliable coverage to all users in overwhelming traffic situations.

Nevertheless, the new service-based structure, including our proposed added feature, has many obstacles holding it from emerging to the current commercial 5G cellular standard. The most irritating and significant concerns for such dynamic service agility are security and user privacy. Moreover, our proposed transition between service providers implicates high-security risks and possible privacy leakages. Therefore, assuming full trust in such heterogeneously managed and operated formation is not realistic. Fake base stations can be easily deployed to take advantage of such overrated trust and enable eavesdropping or massive service disruption. Not to mention the attackers are interested in user tracking and traffic-profiling, allowing them to learn and monitor users’ sensitive information. Additionally, cellular network privacy regulations prevent distributing users’ information and IDs between different service providers, which impedes our proposed inter-cooperation between base stations of different service providers.

To facilitate reliable cooperative service provisioning in presence of intentional or maliciously induced failures and attacks, in this paper, we propose a lightweight privacy-friendly security mechanism for UAV-assisted inter-operator agility. We propose a network model where the supporting UAVs acts as a mediating authenticator between the User Equipment (UE) in need of service and the guest provider’s base station. For this purpose, we adopt a delegation-based digital signature mechanism, or a proxy signature offering a lightweight UAV/UE and UAV/eNB authentication with low message exchange toward the home core network. In proxy signature authentication mechanism, a proxy signer signs a message using a delegated credentials of an original signer [5]. Proxy signature provides data security and user privacy without adding extra communication overhead to the home core network (original signer). The assigned UAVs are authorized by the home network core as its proxy delegated signer for the agility service. For user privacy-preserving purposes, we further assume a concealed user ID for all authentications and communications with the guest network as well as the supporting UAVs. Additionally, to limit collecting user information by the supporting UAV, the only data available for the UAV to use as a concealed user ID and the user’s eligibility indicator to be a part of the inter-operator service. The proposed inter-operator agility cooperative cellular environment framework can be easily integrated under the 3gpp standard for emergency sessions [6].

To this end, the main contributions of this work can be summarized as follows:Enable privacy-friendly seamless inter-operator agility and secure user porting in a cooperative cellular environment.Introducing a novel trustworthy authentication mechanism to facilitate all-party authentication in a zero-trust environment.

The rest of the paper is organized as follows. The related works are summarized in Section 2. Then, the system model distribution and attack models are described in Section 3. Next, the proposed authentication scheme is introduced in Section 4. The experiment setup and evaluation are in Section 5. Then the security and performance analysis are in Section 6. Finally, concluding remarks are provided in Section 7.

## 2. Related Work

In this section, we scan the literature on previous research efforts related to our work.

### 2.1. NextG Architecture

The NextG standard, starting with 5G, provides significant changes toward the existing cellular standards (up to 4G) to support enhanced performance, scalability, and coverage to support millions of users and devices. In addition to the 4G point-to-point based structure, the 5G cellular standard structure is also designed as a service-driven architecture [3,7]. In particular, the new 5G architecture comes with a paradigm change that relies on network slicing and virtual network functions that can be utilized on-demand [8,9].

In addition, the 5G standard offers integration of heterogeneous networks to support enhanced coverage through various types of base stations and relays. One of the considered new portable relays is flying UAVs as a flexible and autonomous operable, handling extra users and maximizing throughput, coverage, and accommodations [10]. The UAVs’ coverage boosting depends on the position selection of the UAVs within the users in need [11,12], which as well plays a critical role in enhancing the 5G network’s overall performance. For instance, in [13], the authors proposed an enhancement of UAVs’ positioning likelihood for optimized cellular coverage utilizing the self-healing neural model.

UAVs function in the 5G network will not be limited to relaying function; they can serve as short-term flying base stations to provide service in certain infrastructure failure circumstances [14]. In [15], the authors investigated a limited functionality drone (i.e., UAV) base station optimal placement in a 5G+ network backhaul, paying regard to the users’ movements effect. In [16], more work was performed on providing adjusting optimal coverage positioning for base station UAVs with varying ray tracing options in different channel situations.

Due to the increasingly demanding streaming traffic anticipated within the NextG network, it is expected to have a down or overwhelmed associated providers’ base station tower with excessive traffic. That is highly expected, especially in highly demanding circumstances such as full stadiums or urgent circumstances such as disastrous situations. Building on the NextG extended dynamic structure capabilities and taking advantage of the UAV mobility; we aim to achieve seamless communication using inter-operator agility in a cooperative cellular environment. We employ UAVs as a mediating point in handling the users’ traffic to an external (guest) provider’s base station to complete the service request without any service interruption. That is an innovative added feature that we propose as a supplement extension to the 5G service-based agility perspective.

### 2.2. 5G Security Standard

The NextG dynamic and highly connected architecture opened the door for several security vulnerabilities, especially with the integration of easily exploited devices such as UAVs. Therefore, an extended security surge is vital to mitigate any malicious attack on the cellular network. The current 5G standard embraces enhanced security protocols that match the new structure to ensure confidentiality, integrity, and authentication [17,18]. The current 5G authentication protocols relies on a service-based architecture (SBA) known as Authentication and Key Agreement (AKA) to accommodate the 5G service-based structure. The AKA protocol enables key agreement and authentication between the subscribers, the Serving Networks (SNs) that have nearby base stations, and Home Networks (HNs) that corresponds to the subscribers’ carriers [19,20].

Additionally, UAVs deployment increases security risks as a vulnerable target due to its limited resources and low-security devices. Nevertheless, any attempt to over equip the UAV with traditional security, and authentication protocols can overwhelm the UAV’s processor and increase battery draining. Therefore, recent UAV security research works are targeting designing lightweight authentication protocols that will not burden the UAVs. For instance, the authors in [21] proposed a two-stage mutual authentication protocol for Software-Defined Network (SDN)-based multi UAV networks operating in surveillance areas.One of the most recent works is [22], the authors proposed a lightweight Physical Unclonable Function (PUF)-based mutual authentication scheme for UAV flying base station in a 5G network. The authentication in [22] achieves secure authentication without the use of a Public Key Infrastructure (PKI) or any time-consuming encryption. However, the scheme assumes a trusted 5G infrastructure and does not take into consideration mutual authentication. In [23], the authors proposed a lightweight privacy-preserving mutual authentication scheme for UAVs and end-user in WiFi ad hoc networks. The proposed authentication scheme in [23] provides a low computation and communication authentication processing while preserving end-user privacy compared to prior works. However, that protocol adds a large number of communication message toward the gateway node, which will add to the backhaul traffic in the case of cellular networks.

Proxy signature as a delegation-based algorithm is also an adequate authentication technique for UAVs lightweight authentication. It can provide data security without bringing any extra computational loads to the UAV. Therefore, some recent work utilized proxy signature as an authentication protocol for limited-resource devices such as sensors and UAVs. There have been multiple works that recite different proxy signature approaches for various purposes [24,25,26]. For instance, in [27], where the authors proposed a blind ID elliptic Elliptic Curve Cryptography (ECC)-based for UAV network based on Mobile Edge Communication (MEC), while the blind proxy signature is an effective digital signature mechanism, it does not contain any information on the signature’s proxy signer (UAV), which increases identity theft attacks [28].

In [29], the authors proposed a low computational cost short certificate proxy signature for UAV authentication to overcome the integrity attacks on vehicular networks. That certificate-based proxy signature is based on Kim, Park, and Won (KPW)’s proxy signature scheme [30], where a proxy key pair depends on the signer’s private key for authentic information on the proxy signer’s identity. Hence, we utilize Kim, Park, and Won’s proxy signature scheme as a part of our digital signature in our mutual authentication protocol.

### 2.3. NextG Users Privacy

Privacy concept in cellular communication is the protection of users’ personal information and access patterns/activities from leakage or exposure. Nevertheless, the cellular communication providers are allowed to share pieces of the users’ activity data with third parties to enhance its end users’ experience without revealing any personalized information. The NextG networks’ highly connected and service-oriented architecture makes preserving the end-users private information more sensitive and tricky, which requires strong privacy policies [31]. Therefore, the 5G introduces stronger data, location, and identity privacy protocols compared to traditional mobile networks.

Recent research focuses on enhancing the 5G privacy aspects that may be exposed during certain 5G new features [32,33]. For instance, in [34], the authors proposed SDN-enabled handover and privacy protection through sharing of user-specific security context information among related access points. In [13], the authors proposed a secure service-oriented authentication framework for IoT services in a 5G network where a privacy-preserving slice selection mechanism is introduced to allow fog nodes to select proper network slices.

The UAVs utilization within the NextG architecture raises a broad range of privacy issues related to UAVs’ data collection and retention, which must be addressed. Accordingly, in [35] privacy design recommendations are proposed, including a Privacy by Design approach that can help ensure that all legitimate interests and objectives are met in a positive-sum. In [36], the authors discuss blockchain-based privacy preservation solutions for 5G-enabled UAV communications.

In our proposed 5G UAV-based inter-operator service agility model, user privacy is critical and can lead to a possible private information leakage. It is a relatively straightforward tactic to catch a user’s International Mobile Subscriber Identity (IMSI) to authenticate a fake user via alternative strategies. One proposed solution assumes roaming agreement between service providers, and hence, instead of authenticating a user based solely on IMSI, a possible consortium blockchain network can be used as a main authenticator [37,38]. Therefore, the process proposed in [37,38] has the IMSI acting as the first step for establishing a handshaking process between networks (a.k.a., a roaming pact). Nevertheless, this blockchain network setup solution reduces the authentication and data transfer speed with small sample sizes. A possible solution to this is known as BigChainDB, which promises to solve problems within the fields of throughput, latency, capacity, querying, permissioning, decentralized control, immutability, and creation and movement of assets [39]. In our proposed privacy-friendly inter-operator agility in a cooperative cellular environment, all the user IDs are concealed and not shared with the guest network as well as the supporting UAVs. Furthermore, the only information collected by the supporting UAV is the concealed user ID and an indicator for whether it is a part of the inter-operator service or not. Such operation protects users’ private information from leaking through the weakest link (i.e., the UAV).

## 3. Preliminaries

### 3.1. UE Authentication in 5G Networks

In this section, we introduce the details about connection establishment process in the 5G AKA protocol. When a new UE is first connected to the 5G cellular network, the UE initializes the authentication process that is managed by the 5G core. The 5G authentication framework supports both 3GPP access and non-3GPP access networks with Extensible Authentication Protocol (EAP) that is also in use for IEEE 802.11 (WiFi) standard [20]. Such a security structure is convenient for IoT devices and UAVs. In this regard, the 5G EAP authentication protocol supports both EAP-Transport Layer Security (TLS) and EAP-AKA protocols, where authentication process is executed between the UE (a client device) and the Authentication Server Function (AUSF)/Unified Data Management (UDM) (i.e., HN) through the Security Anchor Function (SEAF)/Access and Mobility Management Function (AMF) (i.e., SN) as an EAP authenticator [40].

In Figure 1, we provide the details of the AKA security standard. The 5G core network SEAF function triggers the authentication process upon receiving any access-request message from a UE, including the UE 5G-Globally Unique Temporary ID (GUTI) temporary identifier. If the UE is not provided with a 5G-GUTI, a Subscription Concealed Identifier (SUCI) can be used. The SUCI is an encrypted version of the Subscription Permanent Identifier (SUPI) provided to each UE using the public key of the home network, the SUPI should never be sent in plaintext to ensure UE’s privacy. Then, the SEAF sends an authentication request to the AUSF in the core network, which verifies that the serving network request is authorized. If it is a legitimate request, the AUSF proceeds with the authentication procedure by sending an authentication request to the UDM. Next, the Subscription Identifier De-Concealing Function (SIDF) validates the SUCI by decrypting the SUCI and obtains the corresponding SUPI and selects the authentication method configured for the corresponding subscriber, which is 5G-AKA for our case. The UDM then sends an authentication response to the AUSF including an AUTH token, an XRES token, the key $K_{AUSF}$, and the SUPI if not using the 5G-GUTI. $K_{AUSF}$ is the primary authentication key that can be further used to derive subsequent keys for different purposes.

### 3.2. System Model

In this work, we consider a NextG Non-Standalone (NSA) cellular infrastructure (with 4G/5G eNodeBs/gNodeBs towers) where UEs rely on the contemporary 5G Authentication procedure (EAP-AKA or EAP TLS) for connection establishment. We further assume a 4G/5G cellular infrastructure that enables cooperative service provisioning between different service provider networks. The UAVs are assisting with the attachment process to the UE in need of this service, as shown in Figure 2. The UAVs are considered the mediator party between the UEs and the guest network to perform authentication and initiate the connection toward the guest network while preserving user privacy. In this work, UAVs are considered a complementary eNB/gNB support node and not a relay node, and hence, the UAV is considered to be stationary during service-providing communication. We further assume a centralized model where the core network provides the UAVs with the best location of the service with a stable channel connection.

We assume that the UE stores the public keys of the 5G core network during the UE initial attachment and authentication to the home-network cellular provider. Those keys are refreshed and updated securely by the 5G core network periodically. The UEs are pre-registered for the agility services as an extra paid feature for non-emergency circumstances. Nevertheless, all the UEs are directly eligible for the inter-network cooperation services in emergency circumstances. The home core network assigns a Token Key (TTK) to all the users registered for the service, which is refreshed periodically. Those token keys are presented to the assigned UAV to authenticate the UE as a valid user and to show eligibility for the service. The guest network receives a similar key token for authentication purposes and to avoid IMSI or SUCI catchers from posing as a legitimate guest network service provider. The home core network also provides the UE with a set of Concealed Identifications (CIDs) for communication over the guest network for privacy purposes. The CIDs are concealed user IDs to keep the UE identity anonymous to all the parties participating in the data trafficking process (i.e., UAVs and Guest eNB/gNB). This anonymous communication framework preserves the UE privacy during the service porting to a guest network and prevents user profiling attacks.

### 3.3. Threat Model

The considered inter-operator agility cooperative cellular environment is exposed to multiple security and privacy attacks. In this paper, we assume the following threat vectors:Privacy Invasion: The inter-cooperation between different service providers can leak UEs’ information to third parties and/or UE performance monitoring attackers. Additionally, the supporting UAVs as weak IoT devices can be a leaking point of UEs’ information.Malicious UAV: A malicious UAV can capture the communication procedure to maliciously transmit repeated or delayed messages to either the UE or the guest network.IMSI-Catchers Attack: An attacker catches the gNB/eNB IDs and poses as a legitimate guest base station in the network.Compromised UE: An attacker poses a legitimate UE in need of the service aiming to gain access and collect sensitive data.Replay Attack: An attacker uses a sniffing attack on the UAV, UE, and the guest network communication to transmit repeated or delayed messages maliciously.

## 4. Privacy-Preserved Trustworthy Seamless Service Agility

In this section, we propose a connection establishment for secure service porting between service providers to reduce high loads from other service providers in overpowering traffic situations. Such events can accrue due to heavy user activity such as big-game days and sale-days in shopping malls. Such feature can also be a game changer enabling emergency user porting in disaster situations and infrastructure failures or service shutdowns. We propose an authentication delegation drone UAV that offers verified mutual authentication services for UEs in need of connection establishment assistance. UAVs can provide the UEs with the eNB/gNB ID from cooperative service providers that will be able to accommodate it’s connection needs.

We consider the assigned UAV to possess the 5G core network delegation warrant for the assumed *X* service provider and other delegation parameters. The UE in question will send an authentication challenge to the $D_{i}$ UAV asking for the valid signature message, then $D_{i}$ replies with a valid proxy signature message. Once the UE validates the UAV, the UE sends an eNB/gNB ID request to regain communication stability. The UAV then replies with the nearest eNB/gNB ID for the most available service provider. We use public–private key data encryption for communication between the UAV and UE during the Guest eNB/gNB ID assignment in the guest network service porting process. No further communication is required between the UAV and the UE; such an assumption is convenient to both the UAV as a limited resources device and, at the same time, reduce the communication messages between the UAV and UE.

Figure 3 shows the detailed proposed authentication model. In this work, we rely on KPW proxy signature scheme [30], the KPW model is a delegation warrant-based proxy signature where the proxy key pair depends on the proxy signer’s private key. Therefore, the proxy signer’s identity is protected using the node’s authentic key pair $\left(x_{i},y_{i}\right)$. The KPW is considered a robust proxy signature model since it identifies both the original signer’s identity through a Delegation Warrant $w_{i}$ signed by its private key and the proxy signer’s identity through its private key. Once a proxy signer creates its unique proxy signature, it can never repudiate his/her signature as no other node can replicate it.

### 4.1. Delegation Phase

The 5G core network creates a contract document including a specific delegation policy, expiry date, and certificate public key and sign this certificate with the core network private key. This certificate is called a “delegation warrant”, which describes the delegation “message space” or security policy under the delegated authority functions. This delegation warrant is securely uploaded to the supporting UAV, the delegated proxy signer, along with two delegation keys $\left(r_{i},s_{i}\right)$ for delegation warrant signature, as follows,

Let *g* be a generator of a multiplicative subgroup of $Z_{p}^{*}$ with order *p*. Then a random number $k_{i}\in_{R}Z_{p}^{*}$ is selected from this set.The proxy signature delegation keys are generated as follows:
(1)$$\begin{array}{cc} r_{i} & =g^{k_{i}},\\ s_{i} & =x_{c}h\left(w_{i},r_{i}\right)+k_{i}, \end{array}$$
where, $x_{c}$ is the home core network private key and h() is a collision resistant hash function. In addition, as part of this proxy signature, the core network creates a unique delegation warrant $w_{i}$ for each UAV $D_{i}$, as follows:
(2)$$w_{i}=S\left(r_{i},s_{i}\right),$$
where S() is a digital signature function. Note that this delegation warrant is specific to UAV $D_{i}$ because it uses the ($r_{i},s_{i}$) assigned to this UAV.Then, the delegation parameters (i.e., the proxy parameters, the delegation warrant, and the home core network public key $y_{c}$) are assigned securely to the UAV $D_{i}$ as a tuple of $\left(w_{i},r_{i},s_{i},y_{c}\right)$.

### 4.2. Signature Preparation

Once the delegation process ends, the supporting UAV for the UE provider (home network) will prepare the proxy signature keys that will be used to produce the authentication signature as follows,

The UAV $D_{i}$ creates the proxy signature keys, $\left(xp_{i},yp_{i}\right)$, using the delegation tuple from the delegation phase as follows:
(3)$$\begin{array}{cc} xp_{i} & =s_{i}+h\left(w_{i},r_{i}\right)x_{i},\\ yp_{i} & ={\left(y_{c}y_{i}\right)}^{h(w_{i},r_{i})}r_{i}, \end{array}$$
where $\left(y_{i},x_{i}\right)$ are the public-private key pair for the UAV $D_{i}$, respectively.The UAV then creates the following signature:
(4)$$\sigma_{i}=S\left(m,xp_{i},T_{i}\right),$$
where *m* is a random plain text message and $T_{i}$ is a timestamp. Note that since $xp_{i}$ is only known by $D_{i}$, the proxy signature can only be created by a legitimate $D_{i}$.

### 4.3. UAV/UE Mutual Authentication

Once the UE has trouble reaching its home network, it starts pinging its assigned eNB/gNB (home network) by sending a Radio Resource Control (RRC) connection request message. The delegated UAV from the home network intercepts this message as a part of its delegated authority role. The UAV then starts sending its proxy signature message to self-authorize toward the UE and waits for the UE response with its valid concealed ID. The authentications process then follows the following steps,

The UAV $D_{i}$ includes a fresh σ signature to the UE that contains the following signature tuple:
$$\left(\sigma_{i}{|}_{T_{1}},r_{i},w_{i},y_{i}\right)\left|\right|T_{1}.$$The UE validates the signature (i.e., the encrypted service provider delegation warrant ($w_{i}$) using $\left(r_{i},y_{i}\right)$ and compare it to the attached $w_{i}$.The UE sends an encrypted version of the released delegation warrant $w_{i}$ along with the token key provided by the home network core ($TTK_{1}$) and a concealed ID CID as $E_{y_{i}}\left(w_{ACK},CID\right)$, where $E_{y_{i}}$ is an encryption function based on the UAV public key. Furthermore, $w_{ACK}$ is an acknowledgment of the received delegation warrant.The UAV confirms the $T_{1}$ and $TTK_{1}$ in the UE authentication response, then registers the CID of the UE to be used for the agility service.

### 4.4. UAV/Guest eNB Mutual Authentication

Next, the UAV must authenticate itself to the guest service provider, the eNB/gNB guest network, and vice versa using the following Challenge–Response Authentication Process as follows,

The UAV $D_{i}$ sends a service providing request including its proxy signature tuple $\left(\sigma_{i}{|}_{T_{2}},r_{i},w_{i},yp_{i},DID\right)\left|\right|T_{2}$ to the eNB/gNB guest network, where DID is the UAV ID (aka, Drone ID) registered within the guest network by its $y_{i}$.Then, the eNB/gNB validates the signature (i.e., the home network delegation warrant $w_{i}$) using $\left(r_{i},yp_{i},y_{i}\right)$ and $T_{2}$.Next, the eNB/gNB guest network responds with an authentication response containing an encrypted version of the released delegation warrant $w_{ACK}$ along with the home network core token ($TTK_{2}$) and the eNB/gNB guest network ID (eNBID) as $E_{y_{i}}\left(w_{ACK},eNBID\right)$.The UAV confirms $TTK_{2}$, then register the eNBID.

### 4.5. Third-Party Service Porting

The last step in our proposed authentication mechanism is to facilitate the actual porting between the third-party eNB/gNB and the UE requesting the service.

In this part, the UAV sends an attachment request to the eNB/gNB guest network on behalf of the UE requesting the service that includes the encrypted UE concealed ID, $E_{x_{i}}\left(CID\right)$.At the same time, the UAV announces the $E_{x_{i}}\left(eNBID\right)$ (encrypted by its private key) to the UE requesting the service.Once the UE obtains the service provider’s ID, the UE then sends a guest RRC connection request to the third-party eNB/gNB and waits for the eNB/gNB guest RRC response, which establishes the service connection.

## 5. Experiment Evaluation

This section focuses on the experiments conducted to assess the efficiency of the proposed mutual authentication protocol and service-porting mechanism, against the aforementioned attack model.

### 5.1. Experiment Setup

A full 5G open-source implementation platform equipped with UEs, base stations, and core network is yet under development. There have been promising testbeds and platforms that imitate the 5G core network, such as the Free5GC platform [42]. For our experiment, we used srsRAN [43] which is an open-source 4G and 5G software radio library that is able to work with several RF front-end. The initial connection procedure and the bootstrapping signals are identical in both 4G and 5G; therefore, the srsRAN setup is suitable for our experiment.

The UE is equipped with a universal subscriber identity module (SIM card), storing the unique IMSI. The SIM card also stores the corresponding cryptographic keys used in connection initialization that allows the core network to identify and authenticate the UE. Additionally, the UE has its own device-specific unique identity, called the International Mobile Equipment Identity (IMEI), which is also used for identification. These two unique identifiers, IMSI and IMEI are sensitive and can expose the user to several attacks, as aforementioned earlier. Therefore, we do not share these identifiers with external or guest service provider networks; we have replaced those IDs with concealed IDs, warrants, and certificates.

### 5.2. Equipment List

The experiment is conducted using a server having 4 CPUs, Intel(R) Core(TM) i7-7700K 4.2 GHZ. The PC is connected with 4 Universal Software Radio Peripherals (USRPs) and runs VMWare Esxi server hosting 3 Virtual Machines (VMs) as follows,

UAV VM: The UAV virtual machine with 8 cores and 8 GB of RAM is connected to 2 USRP, B205 mini. The first B205 mini USRP runs an srsENB module that accepts connections from home network UEs and provides them with the guest eNB info to port through. Moreover, the second B205 mini USRP runs an srsUE module to communicate with the guest eNBs and facilitate the service port. The VM also runs the home core using the srsEPC module.UE VM: The UE virtual machine with 8 cores and 8 GB of RAM is connected to a B205 mini USRP running our modified version of the srsUE module that supports our proposed extra mode of operation. The extra mode of operation represents the proposed delegation phase in addition to the standard connection procedure.Guest eNB VM: An eNB virtual machine similar to the UE VM, 8 cores and 8 GB of RAM, is also connected to a B205 mini USRP running our modified version of the srsENB module, which accepts additional external connections to support accommodation and porting requests.

We set the UAV B205 mini USRP to change roles or functionality as needed and perform as both srsUE and srsENB. Hence, the UAV is able to execute the mutual authentication process. In particular, the UAV will serve as an srsENB while authenticating the UE and sending the guest eNB info and connection token to the UE. Then the UAV change roles to perform as an srsUE to accommodation request to the guest eNB. Since the two modes of operation, srsUE and srsENB, are not running simultaneously, the most suitable inter-process communication technique is a shared file as a producer-consumer queue. Running separate processes for the srsUE and the srsENB sockets connection can offer the advantages of bi-directional communication but with more significant overhead. However, the extra overhead can be tolerated as the two processes run on two different virtual machines. The experiment setup is shown in Figure 4.

### 5.3. UE-eNB Cellular Network Bootstrapping

We first start our experiment by the bootstrapping procedure of a new UE to be part of the cellular network. The *rrc_procedure* module, as the name suggests, is responsible for the rrc workflow, which manages cell search and selection, connection establishment, configuration to cell handover, and connection reestablishment. However, all the messages are organized as data structures categorized and saved in one of the following modules *ul_ccch_msg*, *ul_dcch_msg*, *dl_ccch_msg*, *dl_dcch_msg*. These modules stand for uplink control messages, uplink data messages, downlink control messages, downlink data messages, respectively.

A new UE starts operation by scanning for the broadcasted *system info block* messages published by nearby eNBs. The new UE starts sensing the base station broadcasting within its operating frequency and, as a default, selects the base station with the highest power.

The next step is initiating the communication toward the eNB via RRC connection. As previously mentioned, the connection workflow is defined in *rrc_procedure*. However, the logic of these functions is described in the *rrc* module located at the srsUE package. On the eNB side, the *rrc* module located at the srsENB package is responsible for parsing and defining the workflow for the different *rrc* messages. The *rrc_ue* module contains the implementation of the other functions related to UE-eNB connection, such as handling connection messages and sending response messages. Then, the experiment message exchange process is described as follows,

The UE initiates a session to the eNB using a Random Radio Network Temporary Identifier (RA-RNTI).eNB assigns a Cell Radio Network Temporary Identifier (C-RNTI) and sends timing adjustment to the UE.Then, the UE adjusts the timing and sends an RRC Connection Request including UE identity using *send_con_request* function from *srsue: rrc* module.Next, the eNB checks on Mobile Management Entity (MME) and Temporary Mobile Subscriber Identity (TMSI) using *handle_rrc_con_req* function from *srsenb: rrc* module then sends the RRC Connection Setup to the UE using *send_connection_setup* function from the same module.The UE acknowledges the message with RRC Connection Complete message using *send_con_setup_complete* function.

### 5.4. UE-UAV Bootstrapping and Mutual Authentication

We next establish the communication set up between the UE and its supporting UAV. we made some changes to the messages’ data structures in the srsRAN source code of the UE to make it able to handle the newly introduced connection modes while maintaining the ability to connect in normal mode. We added a new attribute to specify the connection mode whether it is normal mode or delegation mode, and an identity attribute to the *rrc_conn_setup_s* data structures in *dl_ccch_msg* module. Additionally, adding an identity attribute to the *rrc_conn_setup_complete_r8_ies_s* data structures in the *ul_dcch_msg* module.

The RRC connection establishment between the UE and the UAV is as follows:The UE initiates a session to the UAV using a random RA-RNTI.The UAV, functioning as a flying base station, assigns a C-RNTI and sends timing adjustment to the UE.The UE then adjusts the timing and sends a normal RRC Connection Request since the UE does not recognize whether it is connecting to a normal eNB or the UAV.When the UAV receives a connection request from a valid subscriber, the UAV replies with an RRC Connection Setup message to the UE specifying that the connection is in *Delegation mode* and includes the UAV identity encrypted by the core network’s private key. This identity is pre-loaded to the UAV as aforementioned in Section 4.1.When the UE receives a Connection Setup message with the mode parameter set to *Delegation mode*, it parses the UAV’s identity after validating the proxy signature message. Once verified, the UE sends an RRC Connection Complete message, including an ACK of the UAV’s warrant and the UE CID, while connected to the home core network, this CID is updated and sent periodically to the UE; therefore, the UE can use it in emergency cases.The UAV parses and verifies that the received CID is registered for the inter-operator service agility feature before starting the mutual authentication process with the UE.

### 5.5. UAV-Guest eNB Connection and Mutual Authentication

Next, the UAV starts communicating with the guest eNB to facilitate the privacy-friendly service between the guest eNB and the UE in need of service. The UAV and the guest eNB are required to authenticate mutually, and then the UAV requests accesstoken from the guest eNB, which the UAV sends along with the guest eNB identifiers to the UE.

While communicating with its home network’s subscribers, the UAV communicates with the nearby guest eNBs using its identity, DID, along with its valid proxy signature. As the home core network, the public key is known to the guest eNBs; the UAV DID is easily verified to be authentic. Additionally, the guest eNB uses its identity signed by its home network to verify itself as an authentic eNBs to the UAV. Once each party verifies the other, the UAV and the guest eNB become mutually authenticated.

To achieve this mutual authentication procedure, we had to modify the eNB module in the srsRAN library to handle different workflows. We have three workflows in total, one for the eNB network subscribers, another for mutual authentication with the UAV, and a third that accepts porting requests from the guest UE devices. Any connection request is sent using *send_con_request* function from the *srsue: rrc* module. The *send_con_request* function has *cause* parameter used as an indicator to specify which flow should be followed.

Once the accommodation request is accepted, the guest eNB generates an access token with a specific expiration date stores it as a valid token. Then the guest eNB sends it to the UAV in *accommodation accept* msg defined in the *dl_dcch_msg* module. The accommodation requests data structure contain the *token* and the rrc_transaction_id. Finally, the UAV sends this token to the UE, which will be used later during the service porting request.

### 5.6. Guest Network Service Porting

Once all the parties are authenticated (i.e., UAV-UE and UAV-Guest eNB), the UAV can send the guest eNB info using newly created *tower_info* message passing the *phy_cell_t* data structure and *token* as parameters to the UE. The *phy_cell_t* data structure acts as an identifier for the eNB and the *token* acts as the new concealed identity for this UE. Moreover, the *phy_cell_t* structure contains the pci, earfcn, and cfo_hz parameters, where the pci and earfcn are enough to identify the tower.

The UE starts the service porting connection to the guest eNB by sending connection request using the *send_con_request* function from the *srsue: rrc* module. The *send_con_request* function sends the IMSI/TMSI and the *cause* as identification parameters. Based on the *cause*, the eNB treats the incoming request either as a normal request or porting request. In case of porting request, we specify the cause as *porting* and insert the token instead of the IMSI/TMSI. Supposing the token provided by the UE matches the one generated by the eNB in the accommodation step. In that case, the guest eNB accepts the UE porting request and continues the connection procedure as a regular user without knowing the user’s actual identity. The RRC connection follows the same procedure as the regular RRC connection with one change; the UE uses the token as a concealed ID, CID, rather than its IMSI/TMSI at the connection request.

## 6. Security and Performance Analysis

This section discusses the proposed authentication mechanism’s security analysis and then presents the evaluation results to demonstrate the experiment outcome that validates its efficiency.

### 6.1. Privacy and Security Analysis

The security analysis for our proposed authentication model shows immunity against the following security issues,

*UE Privacy Leak*: All authentication and communication messages toward either the UAV or the guest eNB in our proposed authentication mechanism use a different concealed UE ID, CID, instead of the actual IMEI/SUCI of the UE. That framework protects the UE from being identified, which prevents any leakage of private information. Furthermore, the only information shared with the UAV is whether the UE is a part of the inter-operator service or not. Therefore, since our proposed authentication for the inter-operator service agility feature does not expose any user’s private information during or after handling the traffic to the guest operator base station, our proposed authentication does preserve privacy.*Authentication of Origin*: The delegation phase securely provides the UAVs with an authentic warrant generated and signed by the home network core private key (origin authority), verifying the delegation source during the drone authentication stage. Furthermore, all other authentication messages involve a token key provided by the home network core ($TTK_{i}$) to ensure authenticity. Thus, any UAV proxy signature is based on the home network core, and then the proposed delegation-based scheme proves the authentication of the source.*Authentication of Engaged Components*: The engaged components in the proposed mutual authentication mechanism are the UAV, the UE, and the guest-eNB.The UAV: The proposed proxy signature keys generation is based on the UAV’s (proxy signer) private key. Therefore, the verifier (i.e., the UE and guest-eNB) can identify the proxy signer identity (i.e., the UAV).The UE: As a proxy signature verifier, the UE’s authentication response includes an encrypted version of its concealed ID to verify the destination authenticity.The guest-eNB: As a proxy signature verifier, the guest-eNB accommodation response includes an encrypted version of the guest-eNB ID as verification of destination authenticity.Therefore, the legitimacy of each component in the authentication scheme is mutually satisfied.*Integrity Attacks*: Even if intercepted, the proxy signature messages are undecipherable since it is signed with both the home network’s and the UAV’s private keys. Moreover, all the authentication messages are encrypted with either the UAV’s private or public keys, providing an extra layer of protection to exchanged data between the authentication parties. Therefore, any unauthorized attacker node intercepting the exchanged messages during the authentication or communicating phases will not be able to read, alter, fabricate, or modify the exchanged messages between the authentication parties. Thus, the proposed authentication scheme prevents message integrity.*Replay Attack*: Each proxy signature message originating from a legitimate UAV is timestamped with $T_{i}$ to ensure a fresh signature. Furthermore, all other authentication messages involve a token key provided by the home network core ($TTK_{i}$) with a timestamped nonce to ensure freshness. Therefore, Any illegitimate adversary intercepts and attempts to send a repeated or delayed version of the exchanged messages between the authenticated nodes will fail. Hence, our proposed authentication mechanism is resilient against any replay attack.*IMSI Catchers and Impersonation Attacks*: Our proposed authentication mechanism is designed to protect exchange messages while sharing node IDs by encryption to prevent impersonators from reusing those IDs. The guest-eNB eNBID is accompanied by a token key TTK(i) to prevent accepting services from malicious attackers declaring using fake eNBIDs, i.e., IMSI Catchers. Furthermore, since we proved our proposed authentication scheme’s authenticity of the origin and all engaged components, an impersonation attack is not possible.

### 6.2. Metrics and Baselines

In this set of results, we evaluate our designed authentication solution’s time performance compared to the standard srsRAN package connection procedure. Hence, we have two baselines:**Normal Connection Time**: shows the time taken by the normal *rrc* message connection procedure, including the USIM supporting XOR/Milenage authentication for the UE. This baseline presents time-based compatibility to a standard situation connection setup on the same testing environment.**Authenticated Guest Porting Process Time**: shows the time performance of our proposed UAV-assisted user-agility supported connection setup, including our all parties authentication in Figure 3 from message 1 to 15.

### 6.3. Performance Results

#### 6.3.1. Authentication Delay

We evaluate our designed authenticated privacy-friendly inter-operator agility model performance compared to standard connection procedures under normal conditions. Table 1 shows the time taken by the normal *rrc* message connection procedure between the UE and the home network operator. Table 2 shows the time performance of our proposed modified inter-operator *rrc* messages connection procedure, including the authentication time between all involved parties.

In Table 2, the total inter-operator connection time is divided into four parts as follows,

**Delegation Phase**: The on-land delegation phase preparation time between the home network and the assigned UAV.**UE-UAV**: The connection time of the UE requests the accommodation service from the supporting UAV, including messages 3 through 8 in Figure 3.**UAV-Guest eNB**: The connection time of the UAV requests accommodation from the nearest Guest-eNB participating in the inter-operator agility services represented in Figure 3 in messages 9 through 11.Finally, the last subsection of the table represents the **Guest Network Porting** time, including the ID exchange of the Guest-eNB and the UE via the supporting UAV and the guest network RRC connection setup. The guest network service porting messages exchanged are represented in Figure 3 in messages 12 through 15.

From Table 2, the connection setup total time is under 1 ms which is suitable for cellular communication authentication and connection setup. The UAV-Guest eNB connection time shown in Table 2 can be further optimized if the UAV has two parallel USRPs. In that case, the UAV will use one USRP for connecting with the guest tower so that the connection establishment will occur only once at the beginning of the connection, and then the UAV will ask for a token for each new home network UE requesting the service. In this work, we utilized a simplified system with USRPs and srsLTE package to demonstrate the proposed model efficiency; the total connection setup time will be further optimized in real-time cellular infrastructure.

#### 6.3.2. Authentication Scalability

Here, we evaluate our designed authentication solution’s ability to expand for realistic large networks. Table 3 numerically reveals our delegated-based proposed authentication framework scalability based on the total count of the transmitted and received messages for each entity involved in the authentication (UEs, UAVs, and Guest eNBs). In particular, the authentication scalability test is a pressure qualitative test to our proposed scheme to check the scalability of our framework under an extensive number of UEs requesting data trafficking services at the same time. Hence, this scalability test measures our proposed scheme’s capability of handling more extensive networks.

In Table 3, we increase the number of UEs supported by 1 UAV to show the increasing load effect on the UAV battery Life. The table demonstrates a linear increase in the total transmitted and received messages for both the UAV and the Guest-eNB and an indifferent UE experience. It is worth noting that the proposed authentication does not add any message exchange payload toward the home network core; therefore, it is suitable when no communication is guaranteed toward the home network. Moreover, all the UAV transmitted messages are short-ranged, which is more energy-efficient than reaching the home network core for authentication, and it obsoletes the need for key management.

## 7. Conclusions

In this work, we proposed a novel UAV-assisted user-agility support framework that enables trustworthy seamless service migration in a zero-trust environment. The proposed framework facilitates temporal authentication-authority delegation and proxying to enable preservice, all-party mutual authentication. In our proposed model, the UAV acts as a mediating authenticator between the UE in need of service and the guest provider’s base station. We used bare-metal virtualization to provide the resources needed to host the entire experiment, and 4 UHD devices acted as the main radio interface. The experiments showed that the presented framework facilitated effective and efficient trustworthy service migration between heterogeneous service provider networks. Our future work includes adopting the presented mechanism to secure v2x communications in a highly mobile environment.

## Figures and Tables

**Figure 1 sensors-22-02756-f001:**
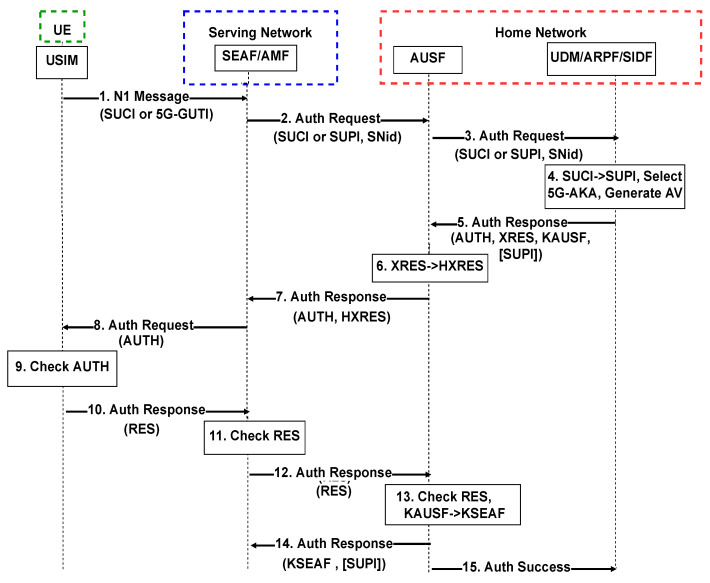
5G-AKA procedure for the UE authentication [41].

**Figure 2 sensors-22-02756-f002:**
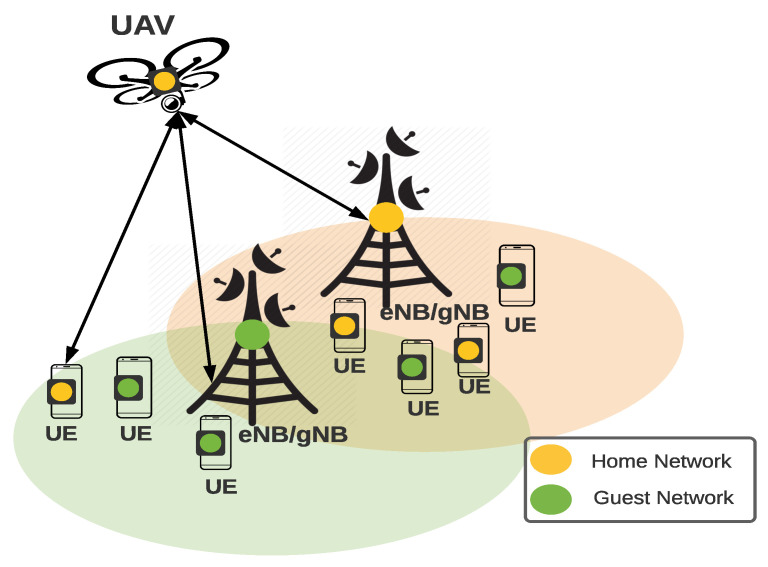
The cooperative service provisioning model.

**Figure 3 sensors-22-02756-f003:**
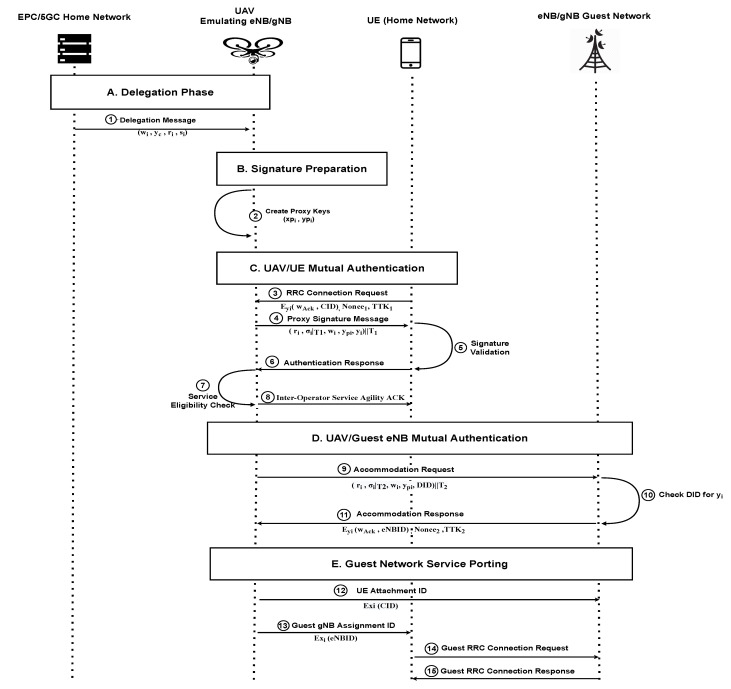
The proposed authentication mechanism exchange messages.

**Figure 4 sensors-22-02756-f004:**
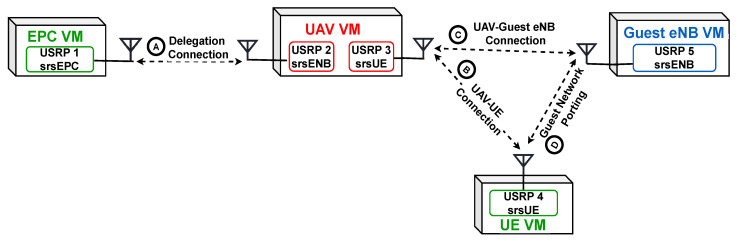
The experiment implementation setup.

**Table 1 sensors-22-02756-t001:** Normal Connection Time Performance.

UE—Home eNB Connection Establishment
0.04 ms

**Table 2 sensors-22-02756-t002:** Authenticated Inter-Operator Connection Performance.

Delegation Phase	UE-UAV	UAV-Guest eNB	Guest Network Porting
0.21 ms	0.06 ms	0.08 ms	0.49 ms
Connection Setup Total Time = 0.84 ms			

**Table 3 sensors-22-02756-t003:** Total number of authentication messages at fixed number of UAVs.

	Home Network				Guest Network	
	UAV		UE		Guest-eNB	
**#** **of UEs**	**TX**	**RX**	**TX**	**RX**	**TX**	**RX**
1	5	3	3	4	2	3
5	21	11	3	4	10	15
10	41	21	3	4	20	30
15	61	31	3	4	30	45
20	81	41	3	4	40	60
40	161	81	3	4	80	120
100	401	201	3	4	200	300

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
