# Peer review of "UAV-Based Privacy-Preserved Trustworthy Seamless Service Agility for NextG Cellular Networks"

_sensors, 2022, doi:10.3390/s22072756_

Round 1

Reviewer 1 Report

In this paper, the authors presents a novel Unmanned Ariel Vehicle (UAV)-assisted user-agility support framework that enables trustworthy seamless service migration in a zero-trust environment.Experiments showed that the presented framework managed to facilitate effective and efficient trustworthy service migration between heterogeneous service provider networks. The work of this paper is practical and logical. However, there are some problems to be further improved as well:

(1)In the system model, this paper mainly aims at the UAV security authentication in 5g network, in order to make readers better understand the article. Therefore, it is necessary to describe the role of some entities in 5gc network.

(2) In the authentication process, the user identity is hidden. It is suggested that the author could explain how to verify CID.

(3)The author has completed the mutual authentication between UAV and UE for the purpose of subsequent communication. It is suggested that the author explain how UAV and UE communicate safely

(4) In introducing the threat Model, for fair comparison, I suggest the author use the "objective and third-party" metrics under realistic adversary models.

"Two Birds with One Stone: Two-Factor Authentication with Security Beyond Conventional Bound." IEEE Trans. on Dependable and Secure Computing, 2018,  15(4): 708-722.  
 "Quantum2FA: Efficient Quantum-Resistant Two-Factor Authentication Scheme for Mobile Devices", IEEE Transactions on Dependable and Secure Computing, 2021

(5)In order to more intuitively show the security of the proposed scheme, I suggest that the author can add formal security analysis.

(6) At present, there are many relevant schemes for UAV privacy authentication. In order to more intuitively display the advantages of the proposed scheme, I suggest that the author can compare with the relevant schemes.

(7)There are some errors in the article that may need to be corrected.For example,"The guest network receives a similar token key for authentication purposes and for avoidingInternational Mobile Subscriber Identity (IMSI) ....."

In all, I suggest a Major revision.

Author Response

Concern # 1:

In the system model, this paper mainly aims at the UAV security authentication in 5g network, in order to make readers better understand the article. Therefore, it is necessary to describe the role of some entities in 5gc network.

Author Response:

We would like to thank the reviewer for the recommendations. Based on this comment we updated our manuscript adding more elaborations on the 5GC network and modules.  In Section 3.1 we emphasized more on the 5GC modules and their operations.

The following paragraph is a sample.

The paragraph is listed  under subsection ``3.1 UE Authentication in 5G Networks’’,

“When a new UE first connects to the 5G cellular network, the UE initializes the authentication process that is managed by the 5G core. The 5G authentication framework supports both 3GPP access and non-3GPP access networks with Extensible Authentication Protocol (EAP) that is also in use for IEEE 802.11 (WiFi) standard. Such a security structure is convenient for IoT devices and UAVs. In this regard, the 5G EAP authentication protocol supports both EAP-Transport Layer Security (TLS) and EAP-AKA protocols, where the authentication process is executed between the UE (a client device) and the Authentication Server Function (AUSF)/Unified Data Management (UDM) (i.e., HN) through the Security Anchor Function (SEAF)/Access & Mobility Management Function (AMF) (i.e., SN) as an EAP authenticator [20,42].”

 Concern # 2:

 In the authentication process, the user identity is hidden. It is suggested that the author could explain how to verify CID.

Author Response:

The UE verifies that the UAV is a legitimate representative from the home network. The process is described in detail in Section 4.3. UAV/UE Mutual Authentication.  Once verified, the supporting UAV checks the token key provided by the home network to the UE to use for porting purposes. The UAV verifies the legitimacy of the service then the home network provides a CID to use during porting to a foreign network. ``The CIDs have concealed user IDs to keep the UE identity anonymous to all the parties participating in the data trafficking process (i.e., UAVs and Guest eNB/gNB). This anonymous communication framework preserves the UE privacy during the service porting to a guest network and prevents user profiling attacks.’’

The explanation above was added to the end of the system model section.

Concern # 3:

The author has completed the mutual authentication between UAV and UE for the purpose of subsequent communication. It is suggested that the author explain how UAV and UE communicate safely.

Author Response:

The details of this process were reviewed and elaborated on in Section 4.5 Third-Party Service Porting, we mentioned the following,

``At the same time, the UAV sends an attachment request to the eNB/gNB guest network on behalf of the UE requesting the service that includes the encrypted UE concealed ID, Exi (CID).'’

We used public-private kay data encryption for communication between the UAV and UE during the Guest eNB/gNB ID assignment in the guest network service porting process. No further communication is required between the UAV and the UE; such an assumption is convenient to both the UAV as a limited resources device and, at the same time, reduce the communication messages between the UAV and UE. We added this explanation to Section 4 for more clarification for the reader.

Concern # 4:

In introducing the threat Model, for fair comparison, I suggest the author use the "objective and third-party" metrics under realistic adversary models.

"Two Birds with One Stone: Two-Factor Authentication with Security Beyond Conventional Bound." IEEE Trans. on Dependable and Secure Computing, 2018,  15(4): 708-722.  
 "Quantum2FA: Efficient Quantum-Resistant Two-Factor Authentication Scheme for Mobile Devices", IEEE Transactions on Dependable and Secure Computing, 2021”

Author Response: We thank the reviewer for the suggested papers. We reviewed the papers and adjusted the manuscript accordingly to add the citation. The work mentioned can also open the door for a future extension that considers 2FA like models in our presented authentication module.

Concern # 5:

In order to more intuitively show the security of the proposed scheme, I suggest that the author can add formal security analysis.

Author Response: Thank you for your suggestion; in this work, we introduce a simple security analysis process to help demonstrate the efficiency of the proposed framework, and its effectiveness to enable privacy-friendly agile service porting in emergency situations. The presented work was also tested on a prototype implementation to ensure its practicality. Given the in-depth design and the detailed implementation process presented in the paper, we preferred to focus our attention on formal security analysis in our sequel paper. We are currently in the process of conducting more formal security analysis and testing the presented system against actual attacks as well. The details of this analysis and any modifications we need to apply to address new challenges will be presented in our sequel papers.

Concern # 6:

At present, there are many relevant schemes for UAV privacy authentication. In order to more intuitively display the advantages of the proposed scheme, I suggest that the author can compare it with the relevant schemes.

Author Response: The primary purpose of this work is to timely implement and validate the proposed new idea of inter-operator agility cooperative cellular environment. We wanted to present our validating experimental work in a realistic 4G/5G implementation environment. Due to the novel nature of the presented framework enabling trustworthy agile service porting, the only valid comparison would be against the standard. Such comparison is presented in a more detailed qualitative discussion in Section  6.3.1. Authentication Delay.

Concern # 7:

There are some errors in the article that may need to be corrected. For example, "The guest network receives a similar token key for authentication purposes and for avoiding International Mobile Subscriber Identity (IMSI) ....."

Author Response: Thank you for your comment, we proofread the paper and checked and corrected all typos and grammatical issues. 

Reviewer 2 Report

The authors have presented a framework that facilitates temporal authentication authority delegation and proxying to enable preservice, all-party mutual authentication. The framework was implemented and tested on top of the srsRAN open-source 4G/5G software stack. They also refer that the framework is for a UAV-Based Privacy-Preserved Trustworthy Seamless Service. However, the "connection" to the UAV is not clear. Because, reliability is critical when a UAV experiences blind spots or
blockages while being stationary or moving aerially. Moreover, transmitting and receiving status updates regarding UAVs’ locations, positions
and conditions under dynamic, unforeseen situations and
environments periodically utilizing Command and Control
(C2) Links is essential. This paper does not consider any of these issues of UAVs. Other corrections and questions are:

-Abstract and Abbreviations. Please correct Ariel -> Aerial

-Abbreviations move it to the end of Introduction section.

-Abbreviations needs to increase more abbreviations, such as: MME, TMSI, UHD, etc.

-Figure 1 There should be a gNB in this figure?

-pp. 222 are directly are -> are directly

-Table 2 Add propagation delay to/from the UAVs

-Table 3 How many UAVs are consideried for 100 UEs?

-There is no single reference with 3GPP documents to check your statements. Please add 3GPP references.

-What would be your proposals to modify 3GPP documents using your framework?

Author Response

Concern # 1:

The authors have presented a framework that facilitates temporal authentication authority delegation and proxying to enable preservice, all-party mutual authentication. The framework was implemented and tested on top of the srsRAN open-source 4G/5G software stack. They also refer that the framework is for a UAV-Based Privacy-Preserved Trustworthy Seamless Service. However, the "connection" to the UAV is not clear. Because reliability is critical when a UAV experiences blind spots or blockages while being stationary or moving aerially.

Author Response:

We thank the reviewer for the suggestion to enhance paper readability. ``In this work, UAVs are considered a complementary eNB/gNB support node and not a relay node, and hence, the UAV is considered to be stationary during service-providing communication. We further assume a centralized model where the core network provides the UAVs a preloaded route, and the location the UAV needs to help with service porting.   

We added the above statement to the paper to clarify this point based on the reviewer's comment.

 Concern # 2:

Moreover, transmitting and receiving status updates regarding UAVs’ locations, positions
and conditions under dynamic, unforeseen situations and environments periodically utilizing Command and Control (C2) Links is essential. This paper does not consider any of these issues of UAVs. 

Author Response: From a centralized cellular network point of view, the UAVs' location and positioning are provided to the serving UAVs through the core network as a preloaded flight and operation map. The UAV is sent to help in emergency situations to help users to port to foreign networks in a privacy preserved manner. Due to the mutual authentication enabled by our system, the UAV is authenticated to the foreign network and can communicate with the home network over IP with no need for long-range ground station control.  More elaborations on this were added to the manuscript in Section 3.2 System Model.

Concern # 3:

Abstract and Abbreviations. Please correct Ariel -> Aerial

-Abbreviations move it to the end of Introduction section.

-Abbreviations needs to increase more abbreviations, such as: MME, TMSI, UHD, etc.

-pp. 222 are directly are -> are directly

Author Response: Thank you for your comment, we proofread the paper and checked and corrected all typos and grammatical issues.

Concern # 4:

-Figure 1 There should be a gNB in this figure?

Author Response: We updated the text on the figure from `eNB`’’ to  ``eNB/gNB’’.

Concern # 5:

-Table 2 Add propagation delay to/from the UAVs

Author Response: The propagation delay is measured in a controlled environment using test equipment.  Due to the many factors affecting this number, and the fact that we build over a full-stack implementation, we used the overall connection establishment time as an indicator of effectiveness. We plan to investigate this issue on a lower level using hybrid SDR modules in our sequel paper. At the current state of implementation,  The isolated propagation delay to/from the UAVs is included as a part of the overall connection establishment time mentioned in the table.  

Concern # 6:

-Table 3 How many UAVs are considered for 100 UEs?

Author Response: The provided text with the table indicates that only 1 UAV is responsible for providing the service for this large number of UEs. That is a pressure qualitative test to our proposed scheme to check the scalability of our framework under an extensive number of UEs requesting data trafficking services at the same time. This scalability test measures our proposed scheme's capability of handling larger networks. This extra explanation was added to the manuscript in Section 6.3.2. Authentication Scalability.

Concern # 7:

- There is no single reference with 3GPP documents to check your statements. Please add 3GPP references.

Author Response: Thank you for your comment, we added the 3gpp standard documentation to our references and mentioned relevant sections within the paper to support our claims and the qualitative discussion.

Concern # 8:

-What would be your proposals to modify 3GPP documents using your framework?

Author Response: The proposed inter-operator agility cooperative cellular environment framework can be integrated under the 3gpp standard for emergency sessions [6]. We are working on a formal proposal to integrate our proposed system framework into the 3gpp emergency sessions.

Reviewer 3 Report

the paper is well structured but needs improvement and development of security theories 

Author Response

Concern # 1:

The paper is well structured but needs improvement and development of security theories 

Author Response: Thank you for your comment, we proofread the paper and checked and corrected all typos and grammatical issues. 

For the development of security theory, in this work, we introduce a simple security analysis process to help demonstrate the efficiency of the proposed framework, and its effectiveness to enable privacy-friendly agile service porting in emergency situations. The presented work was also tested on a prototype implementation to ensure its practicality. Given the in-depth design and the detailed implementation process presented in the paper, we preferred to focus our attention on formal security analysis in our sequel paper. We are currently in the process of conducting more formal security analysis and testing the presented system against actual attacks as well. The details of this analysis and any modifications we need to apply to address new challenges will be presented in our sequel papers.

Round 2

Reviewer 1 Report

The revised version has addressed most of the reviewers' concerns, and I suggest an acceptance. 

Author Response

Reviewer Comment:

The revised version has addressed most of the reviewers' concerns, and I suggest an acceptance. 

Author Response:

We thank the reviewer for his/her feedback.

Reviewer 2 Report

The list of Abbreviations should be written by alphabetical order

Author Response

Reviewer Concern:

The list of Abbreviations should be written by alphabetical order

Author Response:

We thank the reviewer for his/her feedback. The abbreviation is updated to be in alphabetical order.